# Static Postural Control during Single-Leg Stance in Endurance, Team and Combat Athletes from the Spanish National Sport Technification Program

**DOI:** 10.3390/ijerph20054292

**Published:** 2023-02-28

**Authors:** Carlos Villarón-Casales, Arian Ramón Aladro-Gonzalvo, Javier Gámez-Payá, Alberto Pardo-Ibáñez, Fernando Domínguez-Navarro, Diana Gallego, Jorge Alarcón-Jimenez

**Affiliations:** 1Biomechanics & Physiotherapy in Sports (BIOCAPS), Faculty of Health Sciences, Universidad Europea de Valencia, 46010 Valencia, Spain; 2Department of Physical Education and Sport, University of Valencia, 46010 Valencia, Spain; 3Department of Physiotherapy, University of Valencia, 46010 Valencia, Spain; 4Department of Physiotherapy, Universidad Católica de Valencia, 46900 Valencia, Spain

**Keywords:** unipodal balance, sport specialization, young athlete, balance training

## Abstract

In the context of pediatric physical exercise, the analysis of factors affecting postural control (PC) provides insight into the development of sport-specific motor skills. This study aims to evaluate the static PC during single-leg stance in endurance, team and combat athletes from the Spanish National Sport Technification Program. A total of 29 boys and 32 girls, aged 12 to 16 years old, were recruited. Centre of pressure (CoP) was measured on a force platform in standing position for 40 s under two sensorial and leg dominance conditions. Girls showed lower MVeloc (*p* < 0.001), MFreq (*p* > 0.001) and Sway (*p* < 0.001) values than boys in both sensorial conditions (open and closed eyes). The highest values in all PC variables were observed with eyes closed in both genders (*p* < 0.001). Sway values were lower in boys combat-athletes compared to endurance athletes in two sensorial conditions and with non-dominant leg (*p* < 0.05). Young athletes in their teens enrolled in a Sport Technification Program have shown differences in PC when comparing different visual conditions, sport disciplines and gender. This study opens a window to a better understanding of the determinants of PC during single-leg stance as a critical element in the sport specialization of young athletes.

## 1. Introduction

In the context of pediatric physical exercise, postural control (PC) assessment provides insight into the development of sport-specific motor skills [1], has been used in musculoskeletal rehabilitation and injury prevention programs [2,3,4], and has recently been suggested to predict return to play in young athletes with concussion [5,6].

Depending on sport-specific requirements, PC can be evaluated in both static and dynamic positions. While it is true that dynamic PC is a determinant component of an athlete’s physical condition, since most physical activities are dynamic, one might assume that the contribution of static PC is negligible during such activities. Nevertheless, the evaluation of static PC provides useful information for understanding the contribution of different sensory pathways in subsequent motor strategies in order to avoid body imbalance and optimize motor performance [7], mainly of the young athletes.

The two most common tests for assessment of static PC are the timed unipedal stance and monitoring the center of pressure (CoP) movement, while an athlete attempts the single-leg stance motionless on a force platform, either with eyes open or closed [8].

According to the best available evidence, generally, sport practitioners are more skilled than non-sport practitioners [1,8,9,10,11]. Moreover, young female athletes show less body sway than non-athletes and young male athletes [11,12]. Other studies indicate that there are differences between sports and levels of competition [11,12,13,14].

Regarding interlimb postural comparison between adult practitioners and non-sport practitioners, the evidence is contradictory, because the influence of limb dominance on single-leg postural balance is context-dependent [15]. In this regard, the evidence about young athletes is scarce, suggesting no relevant differences between the dominant and non-dominant leg in static unipedal balance tasks [13,16,17].

Currently, it is unclear whether static PC is influenced by anthropometric factors in young athletes. Lee and Lin [18] found worse static one-leg balance in non-athletic males, which could be attributed to the greater weight and the higher inertial moments in comparison to females with open and closed eyes. Furthermore, mesomorphic (muscular) youth had a significantly smaller mean radius of CoP distribution than the endomorphic (fatty) and ectomorphic (linear) youth, in the two visual conditions. Somatotype differences could be due to a relatively higher proportion of musculature in the body, which is beneficial for joint stability and postural control, while a greater height-to weight ratio (increasing the height of center of mass) makes the postural stability more difficult [18]. On the other hand, a recent study shows that the CoP mean velocity of the non-athletes was non-significantly correlated with their age and anthropometric measures under both visual conditions (open and closed eyes) [19]. However, a better anterior-posterior bipodal steadiness of the young artistic gymnasts was associated with a greater age, body height, body mass, and biological maturity (the latter is understood as increased peak height velocity, which is a rapid growth measure). Interestingly, in acrobatic gymnasts, the age, body height or biological maturity were not correlated with anterior-posterior and medial-lateral CoP mean velocities under both visual conditions [19].

Having in mind that single-leg stance is important in sport activities, with differences between sports and levels of competition, and since the dependence of leg dominance on postural ability is not well known, this study aims to evaluate static PC during single-leg stance among endurance, team and combat young athletes from the Spanish National Sport Technification Program. Sport technification is conceived as a previous stage of development and improvement that the young athlete goes through from his/her detection as a talent to an incorporation into high performance sports. The sport technification program promotes and supports the sports training of young athletes: those who have real possibilities of joining different national teams. Our hypotheses were: (i) static PC performance of girls will be better than boys, (ii) the static PC performance of both genders will be worse in the absence of visual information, and (iii) the static PC performance with open and closed eyes will be higher in combat athletes, without differences in leg dominance.

This study represents an important advance in the analysis of postural control in single-leg stance and its differentiation between sport disciplines.

## 2. Materials and Methods

A total of 29 boys and 32 girls, aged 12 to 16 years old, without prior experience in balance assessment participated in the present study. The whole accessible population of the three sport groups was evaluated: 20 endurance athletes (running, triathlon and swimming), 21 team athletes (handball and volleyball) and 20 combat athletes (karate and taekwondo). The procedures were in accordance with the Ethics Standards of the National Committee on Human Experimentation [20], and Declaration of Helsinki. Written, informed consent was obtained from the athletes’ parent or legal tutor after having a full written description of the aims of the study, the possible hazards, discomfort and inconveniences. In addition, all the procedures were verbally explained to the athletes. Inclusion criteria were: (1) regular participation in a sport technification program for at least 4 months, with 5 times per week for 90 to 180 minutes, during both morning and/or afternoon sessions; (2) no diagnosis of cardiovascular, metabolic, orthopaedic, neurological, or vestibular diseases; (3) no history of musculoskeletal injury in the last 2 months prior to the study; (4) normal vision with or without glasses; and (5) no medication affecting the Central Nervous System or known to affect balance and coordination. It was considered an exclusion criterion if one of the parents or legal guardians did not sign the informed consent form, or if the volunteers did not give their consent to participate in the study.

The kinanthropometric assessment was based on the International Society for Advancement of Kineanthropometry protocol [21] and performed by a technician accredited (level 1) on the right-hand side of the body regardless of handedness or stance. Measurements were taken twice, and variation between measures was less than 1% for body mass, height, breadth and girth, and with variability of less than 5% for skinfolds. Participants wore light clothing and were barefoot. Anthropometric measures were the following: (1) height (SECA 225, Birmingham, UK), (2) body mass, was measured to the nearest 0.01 kg using a digital scale (SECA 770, Birmingham, UK), (3) skinfolds (triceps, subscapular, biceps, iliac crest, abdominal, front thigh and medial calf) with calibrated Harpenden callipers (John Bull, British Indicators, West Sussex, UK), (4) girths (arm flexed and tensed, waist, gluteal and calf) were measured using an anthropometric tape (Lufkin W606PM, Cooper Hand Tools, Tyne & Wear, UK), and (5) bone breadths (humerus and femur) were measured with a Holtain anthropometer (Holtain Ltd., Dyfed, UK). The somatotype was estimated by the Heath Carter somatotype method [22]. Body mass index (BMI) was calculated with a technical error of less than 3%.

Prior to the assessment of the balance, the determination of the dominant leg was made by asking the participants which leg would pass or hit a ball [23]. To avoid head movement and reduce the vestibular effect, a reference point (5 cm in diameter) was placed at the height of the participants’ eyes, and 2 m in front of them. A recovery time (60 s) was kept between the performed tests. Participants were not allowed any familiarization trials to avoid learning or the fatigue effect. However, prior to evaluation, a detailed explanation and demonstration of the task was carried out. In addition, participants had the possibility of asking questions about the execution of such procedure. The PC was assessed in silent and standing position for 40 s under the following protocol: (1) dominant leg support with open eyes, (2) non-dominant leg support with open eyes, (3) dominant leg support with eyes closed, and (4) non-dominant leg support with eyes closed.

Vector data of the CoP were measured with Kistler force platform (9253B11, KistlerInstrument AG, Winterthur, Switzerland) at a frequency of 1000 Hz amplified (Max. COP error: ax ≈ 2 mm; ay ≈ 2 mm) and turned from analogical to digital mode with the software Bioware 3.4 (KistlerInstrument AG, Winterthur, Switzerland). The Kistler force platforms are the devices most frequently used in research studies [24,25,26].

Matlab 7.0 (MathWorksRelease 14) program was used to calculate PC variables and conditioning signals. The signals were digitally filtered by using a Butterworth filter fourth order lowpass 0–6 Hz band. The first 10 s of each test were discarded for possible instability. PC variables includes the average speed displacement (Mveloc) of the CoP (mm/s), the rotational frequency (MFreq) and the area per unit time (Sway) (mm^2^/s). The resulting vector (VR) of the CoP displacement was calculated by the following equation:VR = sqrt (AP^2^ + ML^2^)(1)
where AP is anterior–posterior CoP displacement, ML is media-lateral CoP displacement, and sqrt is a mathematical function that returns the square root of a number.

CoP mean velocity has showed good reliability (ICC = 0.75) in the assessment of PC in children and adolescents (6–14 years) [24].

### Statistical Analysis

Normality and homoscedasticity were verified by Shapiro-Wilk and Levene tests, respectively. All data were displayed as means (m) ± standard deviations (SD). In order to study the differences in the somatotype and BMI, separated two-way (gender and group) analysis of variance (ANOVA) were carried out. A two-way (gender, group) ANOVA with repeated measures (sensorial condition and leg dominance) was performed to analyse their effect on the PC dependent variables. Effect size is presented as Partial Eta Squared (η_p_^2^). A value of *p* < 0.05 was considered significant. SPSS 22.0 software (SPSS Inc., Chicago, IL, USA) was used for statistical analysis.

## 3. Results

This section is divided by subheadings. It provides concise and precise descriptions of the experimental results, their interpretation, as well as the experimental conclusions that can be drawn.

### 3.1. Analysis of Anthropometric Variables

Table 1 presents mean and standard deviation values for the anthropometric variables of the endurance, team and combat athletes. The factorial ANOVA showed an interaction effect between somatotype and gender (F_4/48_ = 17.510; *p* < 0.001; η_p_^2^ = 0.593), as well as somatotype and group (F_8/98_ = 3.334; *p* < 0.005; η_p_^2^ = 0.214). In principal effect analysis, there were significant differences between gender and endomorphy (F_1/51_ = 13.693; *p* < 0.005; η_p_^2^ = 0.212), and gender and mesomorphy (F_1/51_ = 6.693; *p* < 0.05; η_p_^2^ = 0.116). Girls exhibited a higher endomorphy component than boys (*p* < 0.001) (M_Girls_ = 4.013 ± 0.217; M_Boys_ = 2.746 ± 0.27), but a lower mesomorphy component (*p* < 0.05) (M_Girls_ = 3.329 ± 0.167; M_Boys_ = 4.123 ± 0.313). On the other hand, significant differences between groups with regard to the endomorphy component (F_2/51_ = 11.286; *p* < 0.001; η_p_^2^ = 0.307) were observed. The differences were established between endurance and combat athletes (*p* < 0.005), and between endurance and team athletes (*p* < 0.001). There were also differences between groups with regard to BMI (F_2/51_ = 3.511; *p* < 0,05; η_p_^2^ = 0.121). Combat had a greater BMI than endurance athletes (*p* < 0.05), on the contrary, combat showed less BMI than team athletes (*p* < 0.05) (Table 1).

### 3.2. Analysis of Postural Balance Variables

Two-way ANOVA with repeated measures revealed an interaction effect between sensorial condition and gender (F_4/51_ = 7.872; *p* < 0.001; η_p_^2^ = 0.382), as well as between sensorial condition and group (F_8/104_ = 2.612; *p* < 0.005; η_p_^2^ = 0.167) in postural balance variables. In the principal effect analysis, there were significant differences between genders with eyes open (F_4/51_ = 7.210; *p* < 0.001; η_p_^2^ = 0.361) and with eyes closed (F_4/51_ = 176.301; *p* < 0.001; η_p_^2^ = 0.933). Post hoc analysis showed that boys had higher values in MVeloc (*p* < 0.001) (M_Girls_ = 34.493 ± 0.965; M_Boys_ = 45.935 ± 1.938), MFreq (*p* < 0.001) (M_Girls_ = 165,637 ± 7.727; M_Boys_= 237.569 ± 12.807) and Sway (*p* < 0.001) (M_Girls_= 257.401 ± 12.061; M_Boys_ = 368.911 ± 19.699) than girls, during open- and closed eyes (Figure 1, Figure 2 and Figure 3). On the other hand, the highest values of all PC variables were observed with eyes closed in both genders (*p* < 0.001) (Table 2).

Likewise, interaction effects were observed between group, gender, and leg dominance (F_4/51_ = 176.301; *p* < 0.001; η_p_^2^ = 0.933). The comparison between endurance and combat athletes showed significant differences in the MFreq (*p* < 0.05) (M_endurance_ = 2.510 ± 0.780; M_Combat_= 2.190 ± 0.550) independent leg dominance and sensorial condition (Table 3). Boys of combat sports showed lower sway values than endurance athletes (*p* < 0.05) (M_endurance_ = 547.025 ± 32.795; M_Combat_= 427.932 ± 32.286) with non-dominant leg in both sensorial conditions (Table 4).

## 4. Discussion

The objective of the present study was to investigate static PC during single-leg stance in young endurance, team and combat athletes from the Spanish National Sport Technification Program. It was expected that (i) static PC performance of girls would be better than boys, (ii) the static PC performance of both genders would be worse in absence of visual information, and (iii) the static PC performance during open- and closed eyes would be higher in combat athletes without differences among leg dominance.

According to the comparison between genders in each PC variable, girls showed lower values than boys with eyes open and closed, results that confirm the first hypothesis. Previous studies have showed similar results [12,13], and the differences between genders can be explained by various factors. First, it may be suggested that static PC performance was conditioned by the maturation processes of the sensory pathways and somatosensory systems. The subjects of the present study are undergoing a maturation process, knowing that this process involves “growing spikes”: peaks affecting their capability to learn a particular motor skill [25]. For instance, in both girls and boys, visual and vestibular afference develops until 15 years of age; however the vestibular system matures earlier in females than in males, at 9 or 10 years of age, and again at 13 or 14 years of age [26]. Nolan et al. [27] found high mediolateral displacement values of CoP among 9 to 10-year-olds, and reduced values among 15 to 16-year-olds, suggesting that at an intermediate age (i.e., 12 to 13 years old), males may still be developing aspects of sensorimotor control. Regarding the proprioceptive system, no gender-related development trend has been observed [26].

Subsequently, the results obtained in the present study suggest that anthropometric factors may explain worse balance outcomes among boys. On the one hand, Lee and Lin [18] found worse static unipedal balance in non-athletic boys, which could be attributed to higher weight and the higher inertial moments in comparison to girls. On the other hand, John et al. [28] reported that an increased peak height velocity was associated with the phenomenon of adolescent awkwardness, a temporary disruption of static and dynamic PC in young male soccer players. It seems that changes in skeletal and muscle-tendon structures during the adolescent growth spurt might disturb the proprioceptive ability and therefore reduce balance control [29]. In our study, boys exhibited a similar ectomorphic component (linear) and BMI than girls (*p* > 0.05). However, there was a trend for boys to be taller than girls in team sports, although this tendency may be attributed to the high height score obtained in one male (as shown by the standard deviation of the mean height). According to Lee and Lin [18], a higher height-to-weight ratio implies greater difficulty for postural stability, as it increases the height of center of mass. Alonso et al. [30] mention that the influence of body height on postural balance is maintained between the genders in adulthood. Additionally, boys were found to exhibit higher mesomorphic (muscular) and lower endomorphic (fatty) component in comparison to girls (*p* < 0.001), which is considered to be beneficial for joint stability and consequently for better PC [15]. Overall, these findings suggest that, while anthropometric factors such as growth may influence PC, the maturation of the visual, vestibular and somatosensory systems may also explain gender differences in balance. Likewise, the study could not determine whether somatotype or BMI can explain the gender difference in PC ability, as a statistical test was not performed with that intention.

Referring to the second hypothesis of the study, it was verified that the static PC performance of boy and girl athletes was worse in the absence of visual information. Maintaining standing postural balance requires an accurate and coordinated interaction between different body segments, the projection of the center of mass falls within the support area, specifically between the feet, and a rapid and continuous feedback from the visual, vestibular and sensory systems [31]. In humans, the visual afference is the predominant sensory pathway to maintain optimal PC [32], the same happens in subjects trained in dynamic balance skills in standing position [33]. Nevertheless, sometimes the success of postural control will depend on the change in sensory weight (from visual to proprioceptive and/or visual to vestibular information, or vice versa) [34]. Studies analyzing balance maturation among the adolescent population have shown that only vestibular input increases when visual and proprioceptive inputs are incongruent [26], just as visual score increases with body height [35]. Finally, the MFreq values were found to be better in combat than endurance athletes, independently of gender, leg dominance and sensorial condition. Sway values with open and closed eyes with non-dominant leg were better in male combat athletes also compared to endurance athletes. These differences could be due to a greater systematic exposure to PC actions among the combat and team athletes. This, in turn, leads to a superior capacity to reduce the influence of imprecise vestibular signals, due to head movements and sudden changes of direction (rotation-translation, acceleration and deceleration), as well as oculo-motor behavior of these athletes (fixations and saccadic eye movements) [36,37]. Another reason for a greater PC in the combat athletes could be explained by muscle strength increasing in the lower extremities. In combat athletes, technical actions demand a significant deployment of muscular strength to surprise or parry the opponent’s attacks. These types of actions require a systematic muscle-strength training that leads to a decrease of the disinhibition and stimulation of muscle spindles, and consequently an improvement in PC [38,39]. On the other hand, when analyzing the dose-response relationship of balance training, it was found that the training modalities (e.g., period and frequency of training) did not have a moderating effect on balance performance in children and youth, but the training intensity was found to be a potential moderator [40].

Further study is required to determine how much of a greater balance ability in young combat athletes is innate and how much is developed through exposure to relevant sports actions, nature, and intensity training. While this study has found differences between athletes participating in different sports, future studies also may analyze the PC performance within the same sport discipline. For instance, balance performance in combat athletes depends on specific control processes and muscle activation patterns, which are different from track and field [41] and team athletes [13]. Despite the fact that none of our athletes had been diagnosed with an orthopedic disorder, we believe that future research also should screen the characteristics of the foot due to the influence that this may have on the stability performance in single-leg standing [1].

The current study present certain limitations that should be mentioned. (1) The analysis of the directional subcomponents of the mean CoP velocity (anterior-posterior and medial-lateral) normalized for the subject’s height was not performed, as the literature recommends [8]. (2) The findings of this study should be considered with caution because of the relatively small sample size in each sport discipline. (3) We evaluated static balance; nonetheless, one-leg balance ability of the analyzed sports is more frequently during dynamic actions. This limitation leads to other studies of dynamic postural control. (4) This study was cross-sectional, and it would be desirable to measure postural control during a one-year training season to confirm our results.

The strength of this study is that it opens a window to a better understanding of the determinants of PC during single-leg stance as a critical element in the sport specialization of young athletes.

## 5. Conclusions

Young athletes in their teens enrolled in a Sport Technification Program have shown differences in PC when comparing different visual conditions, sport disciplines and gender. Girls scored better than boys on all variables recorded with open and closed eyes. In both genders, the static PC performance was worse in the absence of visual information. The sway was less in male combat athletes than male endurance athletes with open and closed eyes and with non-dominant leg. Our findings are useful for coaches who seek to develop specific training based on the balance performance limitations of the ages studied. Limitations of this study are related to the maturation of sensory pathways and the somatosensory system, the changes in the skeletal and muscle-tendon structures during adolescence, as well with the amount of exposure to balance actions, which depend on the sport requirement.

## Figures and Tables

**Figure 1 ijerph-20-04292-f001:**
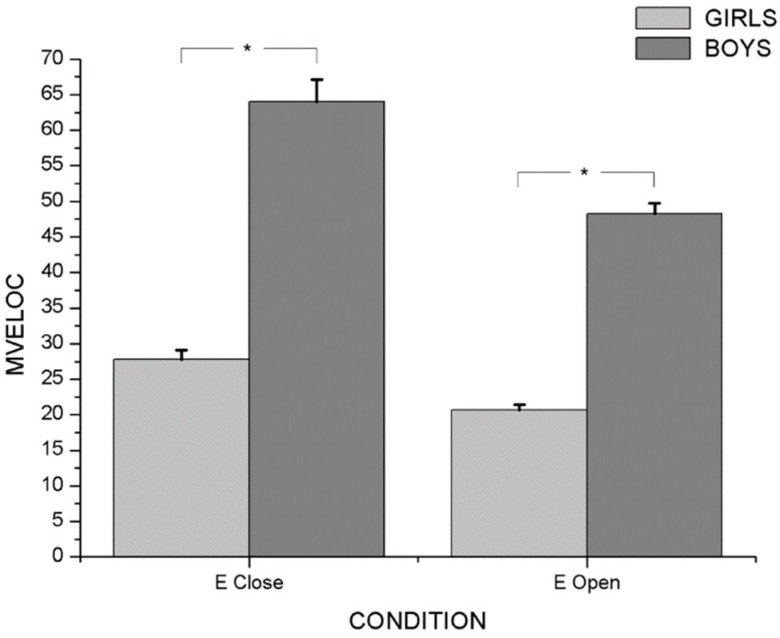
Differences between boys and girls for MVeloc variable with open and closed eyes. * *p* < 0.001.

**Figure 2 ijerph-20-04292-f002:**
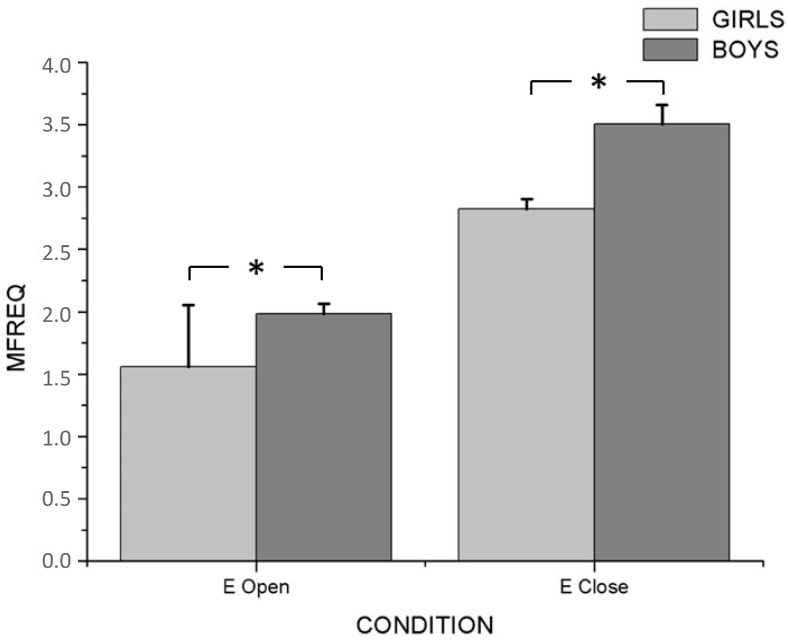
Differences between boys and girls for MFreq variable with open and closed eyes. * *p* < 0.001.

**Figure 3 ijerph-20-04292-f003:**
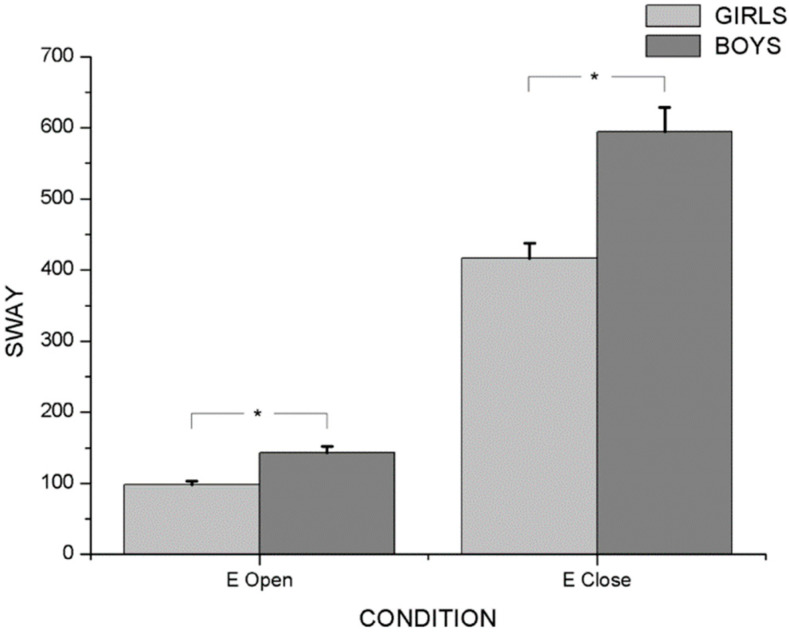
Differences between boys and girls for Sway variable with open and closed eyes. * *p* < 0.001.

**Table 1 ijerph-20-04292-t001:** Characteristics of studied subjects.

	Gender	Endurance Athletes G ^1^ (*n* = 10); B ^2^ (*n* = 10)	Team Athletes G (*n* = 11); B (*n* = 10)	Combat Athletes G (*n* = 11); B (*n* = 9)	Total G (*n* = 32); B (*n* = 29)
Age (years)	Girls	14.63 ± 1.33	15.23 ± 0.85	14.77 ± 0.87	14.90 ± 0.99
	Boys	15.74 ± 1.38	14.76 ± 1.14	14.78 ± 1.51	15.17 ± 1.39
Weight (kg)	Girls	53.47 ± 4.86	61.02 ± 8.96	55.03 ± 7.51	56.70 ± 7.97
	Boys	62.37 ± 9.32	68.9 ± 14.06	58.76 ± 14.47	63.40 ± 12.62
Height (m)	Girls	161.77 ± 3.37	165.67 ± 4.78	148.34 ± 44.78	157.66 ± 29.15
	Boys	144.95 ± 67.30	173.15 ± 9.71	144.84 ± 58.84	153.67 ± 53.37
Endomorphy	Girls	2.92 ± 0.66	4.93 ± 1.24	3.9 ± 0.77	4.01 ± 120
	Boys	1.92 ± 0.26	3.34 ± 1.59	3.16 ± 1.71	2.75 ± 1.41
Mesomorphy	Girls	2.87 ± 0.62	3.77 ± 0.99	3.22 ± 0.93	3.33 ± 0.93
	Boys	3.49 ± 1.13	4.77 ± 1.25	4.2 ± 2.30	4.12 ± 1.59
Ectomorphy	Girls	2.87 ± 0.83	2.36 ± 0.92	2.55 ± 0.69	2.57 ± 0.81
	Boys	3.48 ± 0.75	2.59 ± 1.17	3.13 ± 1.33	3.08 ± 1.10
Body Mass Index (kg/m^2^)	Girls	20.42 ± 1.61	22.15 ± 2.37	20.92 ± 1.17	21.22 ± 1.87
	Boys	20.69 ± 168	22.81 ± 3.37	20.76 ± 3.92	21.44 ± 3.06

^1^ Girls; ^2^ Boys.

**Table 2 ijerph-20-04292-t002:** Results of the principal effect analysis for the sensorial condition and gender.

Variables	Eyes Open		Eyes Closed	
	M	SD	M	SD
MVeloc	34.49	0.97	45.93 *	1.94
MFreq	2.48	0.50	3.80 *	0.84
Sway	257.40	12.06	368.91 *	19.70

* *p* < 0.001.

**Table 3 ijerph-20-04292-t003:** Results of the principal effect analysis for the group.

Variables	Endurance		Combat		Team	
	M	SD	M	SD	M	SD
MVeloc	54.60	13.2	54.40	15.0	53.10	11.80
MFreq	2.51	0.78	2.19 *	0.55	2.50	0.57
Sway	233.0	93.5	281.0	133.30	259.80	125.40

* *p* < 0.05.

**Table 4 ijerph-20-04292-t004:** Results of the post hoc analysis for the group and leg dominance in boys.

Variables	Dominant Leg						Non-Dominant Leg					
	Endurance		Combat		Team		Endurance		Combat		Team	
	M	SD	M	SD	M	SD	M	SD	M	SD	M	SD
MVeloc	59.4	11.0	51.0	13.0	54.0	18.3	59.3	12.6	58.3	16.2	58.5	15.7
MFreq	2.70	0.63	2.23	0.56	2.20	0.79	2.51	0.7	2.16	0.65	2.09	0.59
Sway	223.8	82.9	248.9	119.3	231.0	125.5	547.02	32.79	427.93 *	32.28	304.5	148.1

* *p* < 0.05.

## Data Availability

No new data were created or analyzed in this study. Data sharing is not applicable to this article.

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
