# Peer review of "Static Postural Control during Single-Leg Stance in Endurance, Team and Combat Athletes from the Spanish National Sport Technification Program"

_ijerph, 2023, doi:10.3390/ijerph20054292_

Round 1
Reviewer 1 Report (New Reviewer)
A very interesting article. Methodologically, it meets all the necessary quality requirements. The results and statistical analysis are correct, as is the presentation of tables and figures. The discussion presents very interesting reflections on the topic analysed. And a novel contribution is made on postural balance in a population of young athletes, which needs to be studied.
The authors acknowledge some limitations of the study. We can add a few more. The study is cross-sectional. It would be better to carry out a longitudinal study over at least one year, which would allow us to monitor the subjects' progress. On the other hand, the sample size is relatively small, given that it is divided into several subgroups. It would also have been very interesting to follow the training of the subjects in the sample. Depending on the training they have undergone, this may influence or contaminate the results of the tests. The degree of familiarity of the subjects with the tests should also have been asked.
It is recommended that the authors continue this line of work with further longitudinal research, following up the subjects in the sample and, if possible, increasing the sample size.
Author Response
[A] Dear reviewer, thank you very much for your valuable comments. We hope the changes introduced in the manuscript satisfy your requirements. We also apologize for the previous mistakes in the manuscript, and for the unclear parts of the document. We have included more information in the document to make the manuscript more understandable. We work hard with two main objectives, the first one is to increase the knowledge to stimulate the safe sport actions, as well as to make accessible our small advances to all young athletes when training their postural stability. So, any comment that helps us to improve our purpose is always welcome.
Responses to Reviewer 1
[R1] The authors acknowledge some limitations of the study. We can add a few more.
[R1] The study is cross-sectional. It would be better to carry out a longitudinal study over at least one year, which would allow us to monitor the subjects' progress.
[A] Thank you very much for your suggestion. We have included as follow (page 9 line 573).
“(4) This study was cross-sectional, and it would be desirable to measure postural control during one-year training season to confirm our results.”
[R1] On the other hand, the sample size is relatively small, given that it is divided into several subgroups.
[A] Thank you very much for your comment. This limitation has been discussed as suggest (page 8 line 497):
“(2) The findings of this study should be considered with caution because of the relatively small sample size in each sport discipline.”
[R1] It would also have been very interesting to follow the training of the subjects in the sample. Depending on the training they have undergone, this may influence or contaminate the results of the tests.
[A] Thank you very much for your comment. This limitation has been presented as suggest (page 8 line 484 to 486):
“Further study is required to determine how much of a greater balance ability in combat young athletes is innate and how much is developed through exposure to relevant sports actions, nature, and intensity training.”
[R1] The degree of familiarity of the subjects with the tests should also have been asked.
[A] Thank you very much for your comment. We have mentioned this detail in Materials and Methods section, as follows (page 2 line 159):
“A total of 29 boys and 32 girls, aged 12 to 16 years without prior experience in balance assessment have participated in this study.”

Reviewer 2 Report (New Reviewer)
Postural Balance in Single-Leg Stance the Individual, Collective and Combat Athletes from the Spanish National Sport Technification Program.
The study is interesting. However, I have noticed several inconsistencies which I will briefly outline.
In the title it can be understood: "Posture Balance in Monopodal Support of Individual, Collective and Combat Athletes of the National Sports Technification Program of Spain". There are no individual athletes, but individual and collective sports are practiced.
"open- and closed eyes conditions" = Eyes-Open and Eyes-Closed
"individual, collective and combat athletes" = individual sport, team sport
Regarding the content of the article, it raises the study of single-leg balance in adolescent athletes belonging to the national program of sports modernization in Spain.
It is a study in which it is intended to compare the unipodal balance in three groups of athletes under two conditions (eyes open and closed) and with two types of support (dominant and non-dominant leg). One group practices individual sports, another collective and the last combat sport.
In the introduction, the author mentions: 1) the importance of static postural balance in this population, 2) the way to evaluate it, 3) the aspects that could have an impact on it (anthropometrics/sex); finally, he issues three hypotheses.
From my point of view, a faster and more accurate approach should be made by reorganizing the existing information. Phrases that contribute little to the problem should be eliminated (e.g.: lines 35 to 43 “That is why… to exercise”). In contrast, others could be added to explain the particularity of combat sports in the development of postural balance. It would be convenient to explain why the author chose to compare the type of sports practice (individual/collective) with a discipline (karate and taekwondo) that is part of an individual sport.
Regarding methodology and results, these segments are presented in full. However, it would be pertinent to add the number of subjects per group in the methodology, while a sentence explaining what happened to the non-dominant leg could be included in the results. In addition, the results about the somatotype are included, although a hypothesis was not proposed.
In summary, the article is interesting as there is little literature on the subject addressed. I consider it necessary to restructure the introduction and carry out a complete revision of the English, before submitting it again for revision.
Author Response
[A] Dear reviewer, thank you very much for your valuable comments. We hope the changes introduced in the manuscript satisfy your requirements. We also apologize for the previous mistakes in the manuscript, and for the unclear parts of the document. We have included more information in the document to make the manuscript more understandable. We work hard with two main objectives, the first one is to increase the knowledge to stimulate the safe sport actions, as well as to make accessible our small advances to all young athletes when training their postural stability. So, any comment that helps us to improve our purpose is always welcome.
Responses to Reviewer 2
[R2] In the title it can be understood: "Posture Balance in Monopodal Support of Individual, Collective and Combat Athletes of the National Sports Technification Program of Spain". There are no individual athletes, but individual and collective sports are practiced.
[A] Thank you very much for your comment. We humbly believe that as indexing the term monopodal support is not frequently used, therefore, we propose the following title for you evaluation:
“Static Postural Control during Single-Leg Stance in Endurance, Team and Combat Athletes from the Spanish National Sport Technification Program”
[R2] "open- and closed eyes conditions" = Eyes-Open and Eyes-Closed
[A] Thank you very much for your suggestion. We incorporated changes into the text whenever they were understandable.
[R2] "individual, collective and combat athletes" = individual sport, team sport
[A] Thank you very much for your suggestion. Three groups of athletes are evaluated in this study. For the better understanding of the readers, it has been replaced “individual” by “endurance”; and “collective” by “team” athletes.
[R2] From my point of view, a faster and more accurate approach should be made by reorganizing the existing information. Phrases that contribute little to the problem should be eliminated (e.g.: lines 35 to 43 “That is why… to exercise”). In contrast, others could be added to explain the particularity of combat sports in the development of postural balance. It would be convenient to explain why the author chose to compare the type of sports practice (individual/collective) with a discipline (karate and taekwondo) that is part of an individual sport.
[A] Thank you very much for your comment. We have made the modifications as suggested.
[R2] Regarding methodology and results, these segments are presented in full. However, it would be pertinent to add the number of subjects per group in the methodology,..
[A] Thank you very much for your suggestion. We have included as suggest (page 2 lines 160 to 162):
“It was evaluated whole accessible population of the three sport groups: 20 endurance athletes (running, triathlon and swimming), 21 team athletes (handball and volleyball) and 20 combat athletes (karate and taekwondo).”
[R2] ….while a sentence explaining what happened to the non-dominant leg could be included in the results.
[A] Thank you very much for your suggestion. The effect of leg dominance only occurred in Sway values among combat and endurance athletes (page 6 lines 334 to 339). Leg dominance had no effect in the other analyses or variables.
[R2] In addition, the results about the somatotype are included, although a hypothesis was not proposed.
[A] Thank you very much for your suggestion. Because there is currently insufficient information on the effect of anthropometry on static postural control in young people, we do not present a hypothesis.

Reviewer 3 Report (Previous Reviewer 1)
Thank you for developing your research with the requested corrections.
Author Response
Responses to Reviewer 3
[R3] Thank you for developing your research with the requested corrections.
[A] Dear reviewer, thank you very much for your valuable comments. We hope the changes introduced in the manuscript satisfy your requirements. We also apologize for the previous mistakes in the manuscript, and for the unclear parts of the document. We have included more information in the document to make the manuscript more understandable. Thank you very much for all your suggestions and comments.

Reviewer 4 Report (New Reviewer)
Thank you very much to considering to publish your paper on our Journal. We have some questions that we would want to be clarified:
MATERIAL AND METHODS
1,.- Due to balance performance in the present study , we think that it would be desirable to introduce some "foot characteristics" of the participants´feet; for instance, there is a "foot posture index" to take into account the cavus or the planus foot types, that have a very importante influence on the body balance.
2.- You should include a validation reference of the Posture Balance Board and to include the "standard error of the assessment" of the device.
RESULTS
3.- In order to check the data values obtained, you should add the tables with p value and comparisson results with mean and SD, not only the graphics
DISCUSION
4.- You should to include a short explanation about why your "collective athletes height" is bigger in boys than in girls and perhaps to speak about the center of gravity in them and the differences in balance by this reason.
CONCLUSION
5.- we dont feel the trully goal of the paper: why is important to know the results of the present study? as a lector, I dont find the usefulness of the present paper.
Author Response
[A] Dear reviewer, thank you very much for your valuable comments. We hope the changes introduced in the manuscript satisfy your requirements. We also apologize for the previous mistakes in the manuscript, and for the unclear parts of the document. We have included more information in the document to make the manuscript more understandable. We work hard with two main objectives, the first one is to increase the knowledge to stimulate the safe sport actions, as well as to make accessible our small advances to all young athletes when training their postural stability. So, any comment that helps us to improve our purpose is always welcome.
Responses to Reviewer 4
[R4] MATERIAL AND METHODS
1,.- Due to balance performance in the present study , we think that it would be desirable to introduce some "foot characteristics" of the participants´feet; for instance, there is a "foot posture index" to take into account the cavus or the planus foot types, that have a very importante influence on the body balance.
[A] Thank you very much for your suggestion. As we have mentioned in the second inclusion criterion, our athletes had not been diagnosed with any orthopedic disorder; however, we agree that it is necessary to evaluate the foot characteristics due to their influence on balance. We have included this comment in the discussion as follows (see page 8 lines 490 to 493):
“Despite the fact that none of our athletes had been diagnosed with an orthopedic disorder, we believe that future research also should screen the characteristics of the foot due to the influence that this may have on the stability performance in single-leg standing [5]”.
[R4] 2.- You should include a validation reference of the Posture Balance Board and to include the "standard error of the assessment" of the device.
[A] Thank you very much for your suggestion. We have included, as suggest (page 3 lines 247 to 251):
“Vector data of the CoP was measured with Kistler force platform (9253B11, KistlerInstrument AG, Winterthur, Switzerland) at a frequency of 1000 Hz amplified (Max. COP error: ax ≈2 mm; ay ≈2mm) and turned from analogical to digital mode with the software Bioware 3.4 (KistlerInstrument AG, Winterthur, Switzerland). The Kistler force platforms are the devices most frequently used in research studies [24–26].”
[R4] RESULTS
3.- In order to check the data values obtained, you should add the tables with p value and comparisson results with mean and SD, not only the graphics.
[A] Thank you very much for your suggestion. We have included the tables 2, 3, and 4 in pages 6 and 7.
DISCUSION
[R4] 4.- You should to include a short explanation about why your "collective athletes height" is bigger in boys than in girls and perhaps to speak about the center of gravity in them and the differences in balance by this reason.
[A] Thank you very much for your suggestion. We have included a short explanation in the discussion, as follow (page 7 lines 402 to 404).
“However, there was a trend for boys to be taller than girls in team sports, and this trend can be explained because at least one boy was very tall (as shown by the standard deviation of the mean height).”
[R4] CONCLUSION
5.- we dont feel the trully goal of the paper: why is important to know the results of the present study? as a lector, I dont find the usefulness of the present paper.
[A] Thank you very much for your comment. We have highlighted in the conclusions the usefulness of our study as follows (page 9 lines 584 to 588):
“Our findings are useful for coaches in their intention to generate specific training based on the limitations in balance performance and its relationship with the maturation of sensory pathways and somatosensory system, and the changes in the skeletal and muscle-tendon structures during adolescence, as well with the amount of exposure to balance actions depending on sport requirement.”

Round 2
Reviewer 2 Report (New Reviewer)
An extensive editing of English language and style is required
Author Response
[A] Dear reviewer, thank you very much for your valuable suggestion. We have done an extensive editing of English language and style to make the manuscript more understandable.

This manuscript is a resubmission of an earlier submission. The following is a list of the peer review reports and author responses from that submission.
Round 1
Reviewer 1 Report
Introduction: The introductory part of the study is really well written. The aim and hypotheses of the study are well expressed.
Materials and Methods:
a. How did you determine sample size? plz give details.
b. Exclusion details for volunteers?
c. Did you do any familiarization season?
It would be more correct to use the statistical analysis title in terms of the page layout of the study.
The discussion part of the study is written adequately. The literature was compared with the study and the causes were defined in terms of mechanism. The limitations of the study are mentioned.
Author Response
Thanks for all suggestions.
Minor spell check was performed.
Materials and Methods:
- How did you determine sample size? plz give details.
Reply: We didn´t determine a sample size. All the subjects enrolled in the sport technification program in our study were evaluated and it wasn`t only a sample of the athletes. We have included the correction in the manuscript (line 112).
- Exclusion details for volunteers?
Reply: It was considered exclusion criterion that one the parents or legal tutors did not sign the informed consent, or the volunteers did not provide their assent to participate in the study. We have included the correction in the manuscript (line 126).
- Did you do any familiarization season?
Reply: Participants were not allowed any familiarization trials to avoid learning or fatigue effect. However, prior of the evaluation, we carried out a detailed explanation and demonstration of the task. Besides that, the participants had the possibility of asking questions about the execution of it. We have included the correction in the manuscript (line 148).
It would be more correct to use the statistical analysis title in terms of the page layout of the study.
Reply: We have included it.

Reviewer 2 Report
Authors evaluated youth athletes from spanish national sport tichnification programme in terms of postural balance.
Line 36-44. Sentence is too hard to understand, need to be revised and separated to the parts to be clearer.
Whole introduction needs to be shorten and summarized, I think there is no need to give classical informations, authors dive directly into the topic and explain why there is a need to maket this study ?
I think this study’s aim or results does not related to the talent development and there is no need to mentioned from it in the intro.
Method:
Ethical committe name and approval number has to be added.
How you calculated the number of participants to be include to the study ?
Line 150: add this sentence to the prior to PB measurement.
Line 165: create a separate section as “statistical analysis”.
Line 170: Effect size…
Discussion:
This part also seems no professional and is not related to what you mentioned in the intro.
As a whole, this study needs to be rewrite and resubmit. Manuscript was written careless and lack of reading whole literature in this topic. There is 3 groups of children from various sport branches but no results of comparison within between them in the grapghs. I invite all authors to read the paper and work on it to looks it better and have higher quality.
Author Response
Thanks for all suggestions and comments.
Line 36-44. Sentence is too hard to understand, need to be revised and separated to the parts to be clearer.
Reply: We have included the correction in the manuscript (line 35 to 43).
Whole introduction needs to be shorten and summarized, I think there is no need to give classical informations, authors dive directly into the topic and explain why there is a need to maket this study ?
Reply: We have shortened the introduction. The current state of single leg balance evaluation is shown, and key publications are cited. In addition, the background presented is comprehensible to those scientists working outside the topic of the paper.
I think this study’s aim or results does not related to the talent development and there is no need to mentioned from it in the intro.
Reply: Our intention has been to analyze young athletes (sports talent) with the future potential success in national teams and senior elite sport. This study doesn´t tried to show that variables analyzed are predictors of sports talent.
Method:
Ethical committe name and approval number has to be added.
Reply: All procedures were in accordance with the national legislation about Ethics Standards of the National Committee on Human Experimentation published in Official State Gazette, Law 14/2007, on Biomedical Research; 3 July. https://www.boe.es/buscar/act.php?id=BOE-A-2007-12945 (Content is not available in English).
This law, at the 16th issue mention that, the ethical approval is not required for all biomedical research that does not involve any invasive procedure.
3rd issue.
Definitions.
t) "Invasive procedure": any intervention carried out for research purposes that implies a physical or psychological risk for the affected subject.
Our study was non-interventional and did not involve any invasive procedures. In this sense, all participants were fully informed if the anonymity was assured, why the research was being conducted, how their data will be used and if there were any risks associated. In addition, written, informed consent was obtained from the participants’ parent or legal tutor.
How you calculated the number of participants to be include to the study ?
Reply: We didn´t determine a sample size. All the subjects enrolled in the sport technification program in our study were evaluated and it wasn`t only a sample of the athletes. We have included the correction in the manuscript (line 112).
Line 150: add this sentence to the prior to PB measurement.
Reply: We have given another order to sentences.
Line 165: create a separate section as “statistical analysis”.
Reply: We have included it.
Line 170: Effect size…
Reply: We have included it.
Discussion:
This part also seems no professional and is not related to what you mentioned in the intro.
Reply: We do not understand the term “no professional”. Please give details. The authors have interpreted the results from the perspective of previous studies and the working hypotheses.
As a whole, this study needs to be rewrite and resubmit. Manuscript was written careless and lack of reading whole literature in this topic. There is 3 groups of children from various sport branches but no results of comparison within between them in the grapghs. I invite all authors to read the paper and work on it to looks it better and have higher quality.
Reply: The authors have carefully reviewed the state of art about the topic and have cited the most relevant bibliography. However, we will appreciate receiving from you some more examples of the literature related to this topic. About the second question you ask, the comparison between sports groups was not the purpose of our work, neither the specific hypothesis that were being tested.

Round 2
Reviewer 2 Report
In this condition, still there is not enough scientific quality and writing style is present ın the present paper. Authors have not sufficiently conducted required suggestion by reviewer. I suggest to rewrite whole paper again.